# CXCR3 Expression and Genome-Wide 3′ Splice Site Selection in the TCGA Breast Cancer Cohort

**DOI:** 10.3390/life11080746

**Published:** 2021-07-26

**Authors:** Lauren A. Levesque, Scott Roy, Nicole Salazar

**Affiliations:** Department of Biology, San Francisco State University, San Francisco, CA 94132, USA; llevesque@mail.sfsu.edu

**Keywords:** CXCR3, TCGA, alternative 3′ splice site selection, breast tumor immune infiltration

## Abstract

CXCR3 is a chemokine receptor with two well-characterized isoforms that have unique, context-dependent roles: CXCR3-A and CXCR3-B, which are produced through alternative 3′ splice site selection (A3SS). RNA-seq data from The Cancer Genome Atlas (TCGA) were used to correlate CXCR3 expression with breast cancer progression. This analysis revealed significant CXCR3 expression patterns associated with survival and differential expression between the tumor and adjacent normal tissue. TCGA data were used to estimate abundance of immune cells in breast cancer, which demonstrated the association of CXCR3 with immune infiltration, particularly in the triple-negative subtype. Given the importance of A3SS in CXCR3, genome-wide analysis of A3SS events was performed to identify events that were differentially spliced between breast cancer tissue and adjacent normal tissue. A total of 481 splicing events in 424 genes were found to be differentially spliced. The parent genes of differentially spliced events were enriched in RNA processing and splicing functions, indicating an underappreciated role of A3SS in the integrated splicing network of breast cancer. These results further validated the role of CXCR3 in immune infiltration of tumors, while raising questions about the role of A3SS splicing.

## 1. Introduction

CXCR3 is a chemokine receptor with three known isoforms with unique, context-dependent roles: CXCR3-A, CXCR3-B, and CXCR3-alt. CXCR3-A and CXCR3-B are the best studied isoforms and differ in their 3′ splice site on exon 2, which are 245 bp apart. CXCR3-B uses a more proximal 3′ splice site than CXCR3-A, which results in a longer extracellular N-terminus [1,2] (Figure 1). Notably, CXCR3-B uses a more downstream start codon. However, the mechanism of this alternative translation initiation is unknown. Like CXCR3-A, CXCR3-alt uses the distal 3′ splice site, but interestingly a non-canonical 337 bp intron in the middle of exon 2 is spliced out, resulting in a frame shift and a 5 transmembrane-receptor instead of the classical seven [3]. CXCR3-alt has not been as well characterized. Therefore, in this study, we only focus on CXCR3-A and CXCR3-B.

Available evidence suggests that expression of CXCR3-A leads to increased proliferation and migration, while expression of CXCR3-B inhibits migration and proliferation, while also inducing apoptosis [2]. Immune cells have been shown to mainly express CXCR3-A, with low levels of CXCR3-B co-expression [2,4]. In contrast, endothelial cells and pericytes only express CXCR3-B [1,2].

CXCR3 is highly expressed in Th1 cells, CD8+ T cells, and natural killer (NK) cells [5,6,7]. CXCR3 is essential for infiltration of effector T cells into inflammation sites, including tumors [6]. For example, in a melanoma mouse model, CXCR3 was shown to be necessary for extravasation of CD8+ T cells to the tumor [8]. Similarly, in a mouse model of renal cancer, CXCR3 was shown to be necessary for IL-2 immunotherapy [9]. CXCR3 can also inhibit antitumoral immune activity by mediating migration of regulatory T cells to the tumor, as shown by Redjimi et al. in tissues from ovarian cancer patients [10]. In addition, it was found in mice that CXCR3 is necessary for the migration of NK cells into lymph nodes following dendritic cell stimulation [11]. Although CXCR3 is canonically associated with Th1 polarization, CXCL4 was surprisingly shown to induce Th2 cytokines in CD4+ T cells through CXCR3, and although the isoform was not directly confirmed, it was likely though CXCR3-B [5,6,7,12].

As cancer progresses, tumor cells evolve to express dysregulated chemokine receptor levels, presumably in order to facilitate tumor growth through induction of growth factors, hijacking of the immune system to suppress antitumoral activity, and promoting metastasis [5,13]. These changes appear to also involve changes in CXCR3 isoform expression, as cancer cells switch their ratios to more CXCR3-A and less CXCR3-B. For instance, in clear cell ovarian cancer, it was found that cancerous tissue had lower CXCR-B expression, but the same CXCR3-A expression compared to normal ovarian tissue [14]. Similarly, renal cancer tissue showed increased expression of CXCR3-A and decreased expression of CXCR3-B compared to adjacent normal tissue [15].

CXCR3-A has also been shown to mediate metastasis, such as in a melanoma mouse model where knockout of CXCR3-A via antisense RNA led to a significant decrease in metastasis of tumor cells from the foot pad into the lymph node [16]. Furthermore, migration was shown to be inhibited by an anti-CXCR3 antibody in both ovarian cancer cell lines, SKOV-3 and OVCAR, as well as ovarian cancer samples derived from metastatic sites [17].

Overexpression of CXCR3-B in cancer cells has been shown to inhibit oncogenic activity. For instance, CXCR3-B overexpression in the renal cancer cell line Caki-1 promoted apoptosis through downregulation of the antiapoptotic gene heme oxygenase-1 [18]. Further illustrating this point, in the triple-negative breast cancer cell line, MDA-MB-231, CXCR3-B overexpression inhibited CXCL10 mediated proliferation [19].

CXCR3 also highlights the importance of alternative splicing in cancer given the apparently opposing roles played by CXCR3-A and CXCR3-B. Important roles for alternative splicing in cancer are increasingly appreciated, and multiple large analyses of TCGA data have revealed perturbation of splicing of thousands of genes in cancer, with some splicing-related mutations implicated as drivers of disease [20]. However, comparatively little attention has been paid to the specific form of alternative splicing used in CXCR3-A/B splicing, namely A3SS, with the vast majority of the genome-wide work having focused on exon skipping and to a lesser extent intron retention.

Given the complexity of CXCR3′s roles in cancer, including the very different implications for expression of different isoforms and of expression in different cell types, as well as use of a poorly-studied mechanism of alternative splicing—we sought to better understand the role of CXCR3 in cancer by studying RNA-seq data from the TCGA.

The multiple context-dependent roles of CXCR3 pose difficulties in studying CXCR3 in cancer using bulk tumor tissue, as one does not only have to consider the type of cell expressing CXCR3, but what isoform is being expressed. To understand the role of CXCR3 in cancer, we used bioinformatic techniques to analyze differential CXCR3 expression, at both the isoform and gene levels, as well as CXCR3 splicing regulation.

## 2. Materials and Methods

### 2.1. Data Sources

Exon junction alignment data for all TCGA samples was generated by Kahles et al., 2018 [20]. The data for A3SS events were downloaded from https://gdc.cancer.gov/about-data/publications/PanCanAtlas-Splicing-2018 (accessed on 26 July 2021), with the file name as “merge_graphs_alt_3prime_C2.counts.hdf5”. Gene counts data were accessed through the web portal of TCGA for unique patient tumor samples and adjacent normal tissue for each cohort https://portal.gdc.cancer.gov/ (accessed on 26 July 2021). After filtering for samples of interest, counts were normalized using DESeq2 [21].

### 2.2. CXCR3 Isoform Analysis

From the exon junction count data, the A3SS in CXCR3 was identified and samples were filtered for those that had a sum of at least 20 reads across both proximal and distal junction. To calculate the CXCR3 isoform ratio for each sample, the following equation was used: EJ_A_/(EJ_A_ + EJ_B_), where EJ_A_ is the number of reads aligned to the exon junction used by CXCR3-A and EJ_B_ is the number of reads aligned to the exon junction used by CXCR3-B. For each TCGA cohort that had at least 10 patients, the CXCR3 isoform ratio was correlated with survival. Patients were divided into “High A:B Ratio” and “Low A:B Ratio” groups based on the median ratio. Cox proportional analysis was then performed to determine whether splicing correlates with survival (*p*-value < 0.05). For each TCGA cohort that has at least 5 normal samples in its data set, differential splicing between tumor and normal tissue was analyzed using an unpaired Wilcoxon signed-rank test (*p*-value < 0.05). For each TCGA cohort that has at least 5 patients with both normal and tumor tissue in its data set, differential splicing between tumor and normal tissue was analyzed using a paired Wilcoxon signed-rank test (*p*-value < 0.05).

### 2.3. Differential CXCR3 Gene Expression across Cancer Cohorts

For each TCGA cohort, CXCR3 gene expression was correlated with survival. Patients were divided into “high CXCR3 expression” and “low CXCR3 expression” groups by comparison with the median expression level. Cox proportional analysis was then performed to determine whether splicing correlates with survival (*p*-value < 0.05). For each TCGA cohort that has at least 5 patients with both normal and tumor tissue in its data set, differential CXCR3 expression between tumor and normal tissue was analyzed using a paired Wilcoxon signed-rank test (*p*-value < 0.05).

### 2.4. CXCR3 Subtype Classification

For CXCR3 subtype analyses, CXCR3 expression was stratified based on quartile ranking and shown as high (upper 30th quartile), medium (intermediate quartile), and low (lower 30th) categories. CXCR3 subtypes were then measured for associations with specific demographic or clinical variables by stratifying the population according to these variables (e.g., race and molecular breast cancer subtype).

### 2.5. Estimation of Tumor-Associated Leukocyte Abundance

The CIBERSORT online platform (https://cibersortx.stanford.edu/) (accessed on 26 July 2021) [22] was used to estimate the absolute fractions of 22 leukocyte populations in the BRCA tumor samples based on gene expression estimated as FPKM. The analysis was run with 500 permutations, and quantile normalization disabled (as recommended by the tool for RNA-seq data). Only those samples with a maximum significance value of *p* < 0.05 were included in the final analysis (*n* = 953).

### 2.6. Estimation of T Cell Inflammation

Following the method used in Spranger et al., T cell inflammation was estimated for each tumor based on a defined T cell inflamed gene signature consisting of 158 genes [23]. First, β_i_ coefficients were calculated based on gene expression values using the following formula: β_i_ = (S_i_-μ_i_)/σ_i_ (i = 1,2, … *n*), where μ and σ represent the mean and standard deviation of the ith gene’s expression across all samples, respectively; n is the total number of genes, and S_i_ is the ith gene’s expression in a sample. For each gene, if its coefficient β_i_ > 0.1, then this gene is concluded as ‘upregulated’ in this sample and assigned a score of +1; likewise, if its coefficient β_i_ < −0.1, then this gene is concluded as ‘downregulated’ in this sample and assigned a score of −1; otherwise, this gene is concluded as ‘unchanged’ and assigned a score of 0. Next, the summation of the score across all 158 genes from the signature was computed to estimate T cell inflammation of the tumor sample. As each gene score has values of −1, 0, or +1, the summation of all gene scores ranges from −158 (if all genes are downregulated) to +158 (if all genes are upregulated).

### 2.7. Genome-Wide A3SS Analysis in Breast Cancer

From the exon junction count data, events were filtered for those that had a sum of at least 20 reads across both proximal and distal junction. In addition, analysis was limited to events for which both junctions are annotated in hg19 as accessed through the R package, AnnotationHub, which lead to a total of 5029 events beings analyzed [24]. Then, for each sample and event, percent spliced in (PSI) was calculated using the following equation: EJ_prox_/(EJ_prox_ + EJ_dist_), where EJ_prox_ is the number of reads aligned to the proximal exon junction and EJ_dist_ is the number of reads aligned to the distal exon junction. For each A3SS event, differential splicing between tumor and normal tissue was analyzed. Significance was determined using a paired Wilcoxon signed-rank test (FDR < 0.05). Functional enrichment analysis was performed based on the parent genes of significant A3SS genes. The R package, ClusterProfiler was used to test for significant enrichment in GO and KEGG terms (FDR < 0.05) [25].

## 3. Results

### 3.1. CXCR3 Differences in Expression but Not Splicing between Tumor and Healthy Tissues

Given CXCR3′s opposing roles in tumors, both due to differences in isoform function and differences in impact of expression in different cell types (e.g., proliferating cells versus immune cells), we studied overall isoform usage in bulk samples from The Cancer Genome Atlas (TCGA). Using standard methods, we found no significant correlation between CXCR3 splicing and survival, nor was there any differential CXCR3 splicing detected between tumor and adjacent normal tissue (Table 1) in the breast cancer (BRCA) TCGA cohort. It is possible that we would have detected differences in a larger sample set and deeper sequencing sample, since many samples had few reads mapping to the splicing event and thus did not pass the filtering criteria (sum of ≥20 reads across both exon junctions).

Next, we studied total transcript levels of CXCR3 in TCGA samples. The breast cancer (BRCA) TCGA cohort analyzed showed high CXCR3 expression associating with better survival (Table 2, Figure 2A; hazard ratio = 1.4, *p*-value = 0.04). In addition, the BRCA cohort had significantly different CXCR3 expression between the tumor and adjacent normal tissue, showing CXCR3 upregulation in the tumor (Table 2, Figure 2B).

### 3.2. High Levels of CXCR3 Expression in Triple-Negative Breast Cancer

Truly understanding gene expression in breast cancer requires accounting for heterogeneity of disease across patients. Of particular importance is triple-negative breast cancer (TNBC), classified as tumors that do not express the estrogen receptor (ER), the progesterone receptor (PR), or the human epidermal growth factor receptor 2 (HER2) [26,27]. TNBC is associated with a worse prognosis compared to other breast cancers and is more frequent among African Americans and young women [26]. Among patients who received adjuvant therapy, those with TNBC had a worse prognosis than those with Luminal A breast cancer, which express ER and/or PR, but not HER2 [27]; however, when analyzing patients who only received surgery and no adjuvant therapy, there was no difference in outcome between the subtypes. This indicates that TNBC is not necessarily more deadly, but that current therapies are inadequate [26]. Immunotherapy shows promise to be particularly helpful for treating TNBC, as these tumors show a higher abundance of immune infiltration due to increased presence of neoantigens produced by a high mutation rate [28]. In addition, there is a large body of evidence that associates increased immune infiltration of TNBC tumors with better response to adjuvant therapy [28]. It is thus critical to understand the molecular mechanisms of immune infiltration in TNBC to develop optimal therapies. Given that, understanding the relationship between CXCR3 expression, molecular subtype, and immune infiltration could be of particular value to treating TNBC.

Analysis of CXCR3 expression across the molecular subtypes revealed increased CXCR3 expression in HER2-enriched and triple-negative breast cancer subtypes, compared to the luminal subtypes (Figure 3A). Interestingly, when looking at all BRCA tumor samples for differential CXCR3 expression between races, Black BRCA patients show increased CXCR3 expression compared to white patients (Figure 3B). However, the same analysis performed for just TNBC samples shows no differential CXCR3 expression between the races (Figure 3C). This finding indicates that the increased CXCR3 expression in Black patients is most likely due to a higher proportion of Black patients with TNBC compared to white patients (21.8% vs. 13.4%).

### 3.3. High Levels of Leukocyte Infiltration in Triple-Negative Breast Cancer

Given this perplexing pattern, in which CXCR3 was significantly upregulated in the breast cancer tumors, but also associated with better survival, we hypothesized that CXCR3 expression was associated with an antitumoral immune response. To test this hypothesis, we sought to understand cell type abundance in the samples.

We estimated immune cell type abundance for each sample using CIBERSORT. Comparison across samples which showed that CXCR3 expression positively correlates with the abundance of tumor-associated leukocytes (TAL) (Figure 4A). In addition, when comparing TAL abundance between molecular subtypes, TNBC samples show the highest leukocyte abundance, while Luminal B samples (express PR and/or ER, as well as HER2) show the lowest (Figure 4B), which corresponds to differential CXCR3 expression between molecular subtypes (Figure 3A).

Analysis of individual leukocyte populations shows high CXCR3 expressing tumors have increased abundance of macrophages, naïve B cells, and various types of T cells, while having decreased abundance of eosinophils, activated mast cells, and neutrophils (Figure 5A). TNBC samples show high abundance of many types of leukocytes including macrophages (M1 and M0), T cells (CD4 memory activated and follicular helper), and dendritic cells (Figure 5B). Interestingly TNBC samples show the lowest abundance of resting mast cells, but the highest abundance of activated mast cells, indicating that the specific tumor microenvironment of TNBC tumors somehow activate mast cells.

To further investigate the relationship between CXCR3 expression, molecular subtype, and immune infiltration in BRCA tumors, T cell inflammation was estimated for each tumor based on a defined T cell inflamed gene signature consisting of 158 genes. Based on this estimation CXCR3 expression was found to positively correlate with increased T cell inflammation (Figure 6A). In addition, TNBC and HER2-enriched tumors showed increased T cell inflammation compared to the luminal tumors (Figure 6B), corresponding to differential CXCR3 expression between molecular subtypes (Figure 3A).

### 3.4. Significance of A3SS in RNA Processing Genes in Breast Cancer

Given that CXCR3 alternative splicing is among the clearest single alternative splicing events with effects on cancer, it is perhaps surprising that the mechanism of CXCR3 alternative splicing, A3SS, remains poorly studied, with most genome-wide analyses of splicing focusing on exon skipping. We used RNA-seq data from TCGA to estimate percent spliced in (PSI) for each annotated A3SS event in the human genome to identify A3SS events that have differential splicing in tumor and normal tissues in breast cancer patients.

Among the 5029 A3SS events analyzed, 481 events were detected as differentially spliced between breast tumor tissue and adjacent normal breast tissue, with 266 events showing decreased PSI in the tumor and the other 215 events showing increased PSI in the tumor (Figure 7A). Based on the significant A3SS events, the samples were well separated into normal and tumor groups by unsupervised hierarchical clustering (Figure 7B), except for a handful of samples. 

The events that showed the most decreased PSI in the tumor were those in ANKRD36 (an uncharacterized gene with an ankyrin repeat domain), LMO7 (PDZ and the LIM domain-containing), and NKTR (facilitates natural killer cell binding to targets). Interestingly, expression of NKTR is regulated by IL-2 through alternate splicing [29]. LMO7 has recently been implicated in cancer, showing it promotes pancreatic cancer progression, but is associated with better prognosis in prostate cancer [30], p. 7, [31]. The events that showed the most increased PSI in the tumor were those in TPM1 (member of the tropomyosin family), COL5A1 (encodes alpha chain for fibrillar collagen), DEPDC5 (inhibits mTORC1 pathway), PIFO (involved in cilia formation), SREK1 (SR splicing protein) and SNRPD1 (part of the spliceosome). The use of the proximal 3′ss in SREK1 leads to a premature terminating codon (PTC) (Figure 8A), which would most likely mean an unstable transcript or protein and decreased SREK1 activity in the tumor would be predicted. In contrast, SNRPD1 transcripts that use the distal 3′ss have a PTC (Figure 8B), again leading to an unstable transcript or protein and thus increased SNRPD1 activity in the tumor would be predicted. These predictions would need to be validated by studying expression at the protein level, or by performing RNA-seq following inhibition of the NMD pathway; both of which are beyond the scope of this project.

As shown in Figure 9, specific GO terms closely related to RNA splicing were significantly enriched by the parent genes of significant A3SS events for both changes in splicing. There were no significantly enriched KEGG pathways by the parent genes of significant A3SS events for either change in splicing, and there were no molecular function GO terms enriched by parent genes of significant A3SS events that had decreased splicing in the tumor.

## 4. Discussion

### 4.1. Cell Type and Splicing in Cancer

Despite the growth of single-cell RNA-seq techniques, the vast majority of RNA-seq studies still focus on total gene transcript levels in bulk samples. For samples as complex as tumors, these approaches may overlook various important axes of gene expression, two of which are highlighted by our results. First, we show that accounting for cell type abundance clarifies the role of CXCR3 expression in tumors, with immune cell infiltration explaining patterns of gross CXCR3 expression and revealing differences in molecular type across patient tumors. Second, by focusing on isoform usage, we reveal evidence for differential splicing regulation for hundreds of genes in cancer.

### 4.2. Immune Cell Abundance Predicts CXCR3 Expression: Implication for Immunotherapy

We report a seemingly paradoxical pattern of expression of CXCR3 expression, in which elevated CXCR3 expression is associated with tumors relative to healthy tissue, but is also associated with increased patient survival. The correlation of high CXCR3 expression and better survival found in BRCA could be due to increased CXCR3-B expression within cancer cells. However, this trend is most likely due to increased abundance of CXCR3-A expressing immune cells, since among cell types, immune cells tend to have the highest CXCR3 expression (of either isoform). This hypothesis is further supported by the increased expression of CXCR3 in BRCA tumors compared to adjacent normal tissue, indicating CXCR3 is involved in a tumor specific immune response.

The results indicate that triple-negative breast cancer samples show transcriptomic profiles consistent with increased lymphocyte infiltration compared to other subtypes and identify CXCR3 as a potential mediator of that increased recruitment. This second hypothesis is further supported by Mikucki et al., who found in a mouse model of melanoma that CXCR3 expression on CD8+ T cells is critical for its extravasation to the tumor [8]. CXCR3 shows a positive correlation with memory CD4 T cells, but a negative correlation with naïve CD4 T cells, which is in line with the observation that CXCR3 is absent in naïve T cells but is then rapidly upregulated following dendritic cell induced activation [6]. A similar relationship is found between CXCR3 and NK cells, where CXCR3 positively correlates with activated NK cells and negatively correlates with resting NK cells, which again suggests that CXCR3 is involved in NK cell activation.

Some limitations to this study include the inability to determine which cells are expressing CXCR3. While it is possible that the cancer cells are responsible for most of the expression which then attracts immune cells, it seems more likely that the CXCR3 expression derives mostly from the immune cells themselves. In addition, we do not know the ratio of CXCR3-A to CXCR3-B, which can also impact the mechanism of immune infiltration.

Our failure to find evidence for altered CXCR3 splicing between tumors and healthy controls seems initially surprising given previous reports of differential expression in cancers. However, the insignificant CXCR3 isoform finding is most likely due to the low number of reads aligned to either exon junction in CXCR3, thus decreasing our statistical power. Due to the generally low expression of CXCR3 in the tumor and that alternatively spliced gene region is so small, it will be necessary to use specific CXCR3 targeting techniques (such as qPCR or a Western Blot) to fully analyze its splicing.

Although further investigation is needed, these results have implications for the development of immunotherapies for breast cancer. A current challenge for immunotherapies is increasing immune infiltration of the tumor and exploiting CXCR3 mediated migration is a promising avenue towards improving TAL-dependent immunotherapies [32]. This is especially relevant for those with TNBC, where a clear positive relationship between increased immune infiltration and better prognosis has been established, and there is a desperate need for better therapies for this cancer.

### 4.3. Underappreciated Alternative 3′ Splice Site Usage Illuminates Cancer-Associated Splicing Networks

We report the first genome-wide study of alternative 3′ splice site usage in cancer, highlighting hundreds of events that are differentially spliced in breast cancer. Genes with differential A3SS splicing are themselves enriched in genes involved in splicing and RNA processing, a finding that supports the notion that these events contribute to an intricately cross-regulated gene expression regulation network. While alternative splice site usage has often been treated as largely a mere curiosity relative to the phenomenon of exon skipping, these results indicate the importance of better understanding the regulation and implications of alternative splice site usage. A key priority going forward will be understanding the factors regulating the differential splicing events identified here.

The enrichment analysis of events that had a similar predicted splicing pattern to CXCR3 (decreased PSI in tumor), did not reveal any functions that differential CXCR3 splicing is known to affect. However, this evidence did reveal significance of A3SS in the global splicing network in breast cancer. It is well known that proteins involved in splicing often autoregulate their expression by promoting splicing that creates PTCs in their transcripts [33]. Given this, it is not surprising that terms related to RNA splicing were enriched in parent genes of significant A3SS genes. In addition, some of the top A3SS events were in genes that code for splicing proteins (SREK1 and SNRPD1) which can result in PTC-containing transcripts. This evidence suggests a complex splicing network which changes in breast cancer, and is potentially mediated through A3SS.

## 5. Conclusions

Although there were not enough data to analyze CXCR3 isoform expression across the BRCA TCGA cohort, from an additional pan-cancer analysis of 32 TCGA cohorts we analyzed, across all cancers, there was no significant correlation between CXCR3 splicing and survival, nor was there any differential CXCR3 splicing detected between tumor and adjacent normal tissue (Appendix A). Like in BRCA, some TCGA cohorts and analyses at the gene level did reveal divergent CXCR3 patterns among cancers. In some cancers, there was increased CXCR3 expression in the tumors, while in others there was decreased CXCR3 expression (Appendix A). Similarly, high CXCR3 expression was associated with better survival in some cancers and worse prognosis in others (Appendix A). These differing results underlie the importance of understanding the context of CXCR3 expression to predict oncogenic effects.

In the breast cancer cohort, CXCR3 was significantly upregulated in the tumor, and associated with better survival. While at first glance this might seem contradictory, previous research has shown CXCR3 is associated with a tumor specific immune response, which could explain these findings. Further analysis of CXCR3 expression in the breast cancer cohort demonstrated its role in mediating infiltration of immune cells (particularly T cells) to the tumor. Following this trend, triple-negative breast cancer samples showed both high levels of immune infiltration and CXCR3 expression compared to the other breast cancer molecular subtypes. These findings highlight the importance of CXCR3 in immune infiltration, which can be exploited to develop better immune therapies, particularly for triple-negative breast cancer.

Genome-wide A3SS analysis in breast cancer revealed its underappreciated role in the splicing network of breast cancer. There was an enrichment of genes involved in RNA splicing and other post-transcriptional modifications that showed differential splicing in their 3′ss. Furthermore, two genes involved in splicing (SREK1 and SNRPD1) had some of the most differentially spliced A3SS events, consistent with the importance of autoregulated splicing networks in cancer.

## Figures and Tables

**Figure 1 life-11-00746-f001:**
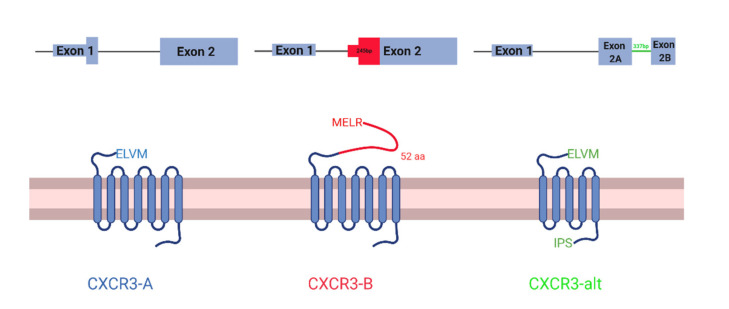
Schematic of CXCR3 isoforms. CXCR3-A (left), CXCR3-B (middle), and CXCR3-alt (right). mRNA transcripts at the top and protein structure at the bottom. In mRNA transcripts, thin lines represent the intron, thick rectangles represent the protein-coding region of the exons, and thin rectangles represent the non-protein-coding regions of the exons.

**Figure 2 life-11-00746-f002:**
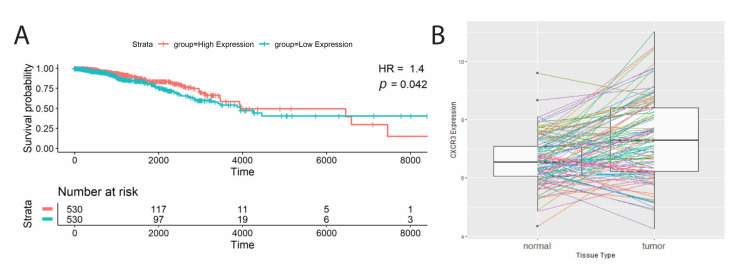
Significant survival correlations with CXCR3 gene expression. (**A**) Kaplan–Meier survival curves of patients split at median CXCR3 expression into two groups, low(blue) and high (red), where high CXCR3 expression was correlated with better survival (Cox proportional analysis, *p*-value < 0.05, HR > 1). (**B**) Significant differential CXCR3 gene expression between tumor and adjacent normal tissue. BRCA cohort shows increased CXCR3 expression in tumor tissue (paired Wilcoxon signed-rank test, *p*-value < 0.05).

**Figure 3 life-11-00746-f003:**
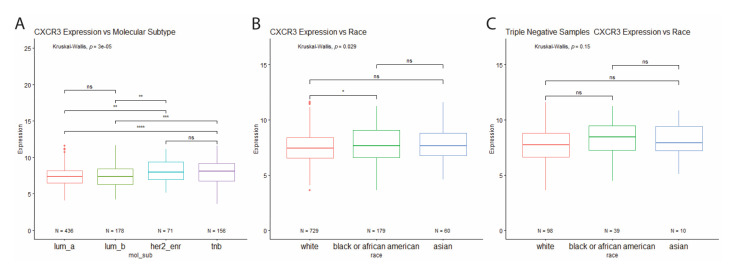
(**A**) Differential CXCR3 expression between molecular subtypes. Increased CXCR3 expression in HER2-enriched and TNB tumor samples compared to luminal tumor samples. * *p* < 0.05, ** *p* < 0.01, *** *p* < 0.001, **** *p* < 0.00001, and ns = not significant. (**B**) Differential CXCR3 expression between races. (**A**) Increased CXCR3 expression in all BRCA tumor samples from Black patients compared to white patients. (**C**) No differential CXCR3 expression between races in triple-negative samples. * *p* < 0.05, ** *p* < 0.01, *** *p* < 0.001, **** *p* < 0.00001, and ns = not significant.

**Figure 4 life-11-00746-f004:**
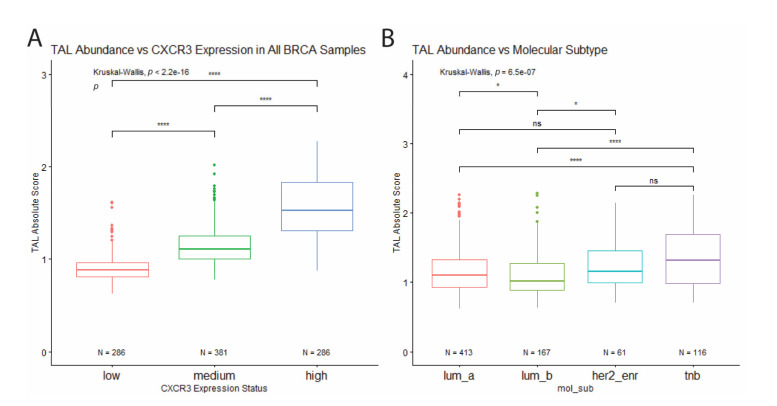
Correlation of total leukocyte abundance with CXCR3 expression and molecular subtype. (**A**) TAL absolute score positively correlates with CXCR3 expression. (**B**) Among the molecular subtypes, TNB samples show the highest leukocyte abundance, while Luminal B samples show the lowest. * *p* < 0.05, ** *p* < 0.01, *** *p* < 0.001, **** *p* < 0.00001, and ns = not significant.

**Figure 5 life-11-00746-f005:**
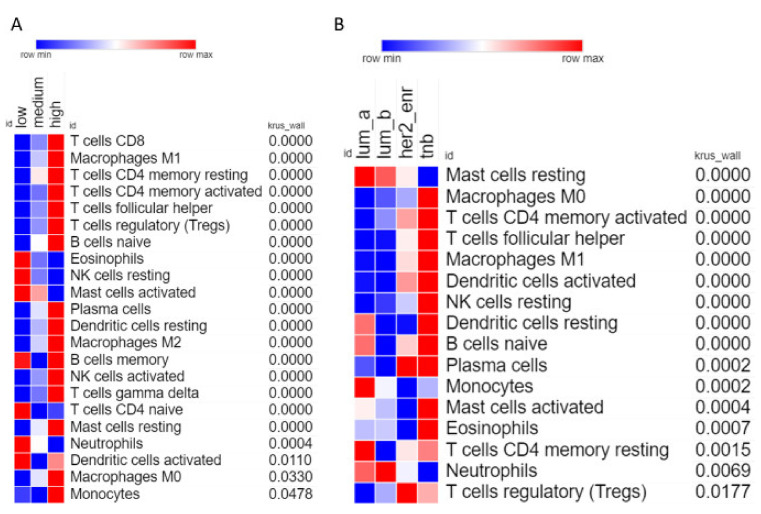
Correlation of individual leukocyte populations with CXCR3 expression and molecular subtype. (**A**) Heatmap showing abundance of each leukocyte for low, medium, and high expressing CXCR3 tumor samples. Only includes leukocytes where the Kruskal–Wallis test had *p*-value < 0.05. Leukocytes are ordered based on *p*-value, most significant at the top, least significant at the bottom. (**B**) Same as (**A**), but for molecular subtypes.

**Figure 6 life-11-00746-f006:**
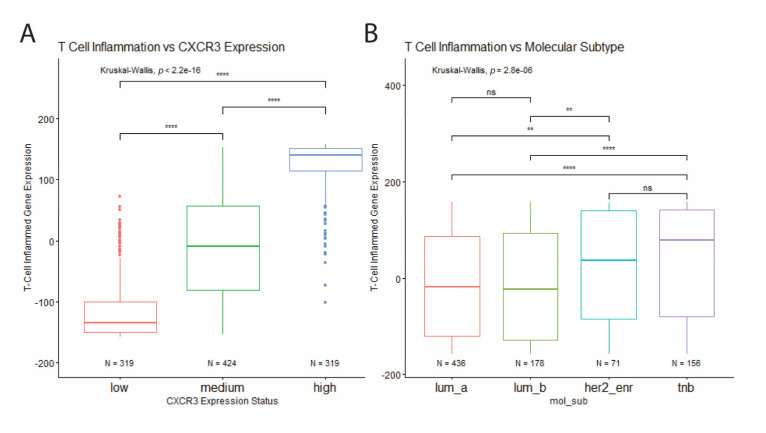
Correlation of T cell inflammation with CXCR3 expression and molecular subtype. (**A**) T cell inflammation positively correlates with CXCR3 expression. (**B**) Increased T cell inflammation in HER2-enriched and TNB tumor samples compared to luminal tumor samples. * *p* < 0.05, ** *p* < 0.01, *** *p* < 0.001, **** *p* < 0.00001, and ns = not significant.

**Figure 7 life-11-00746-f007:**
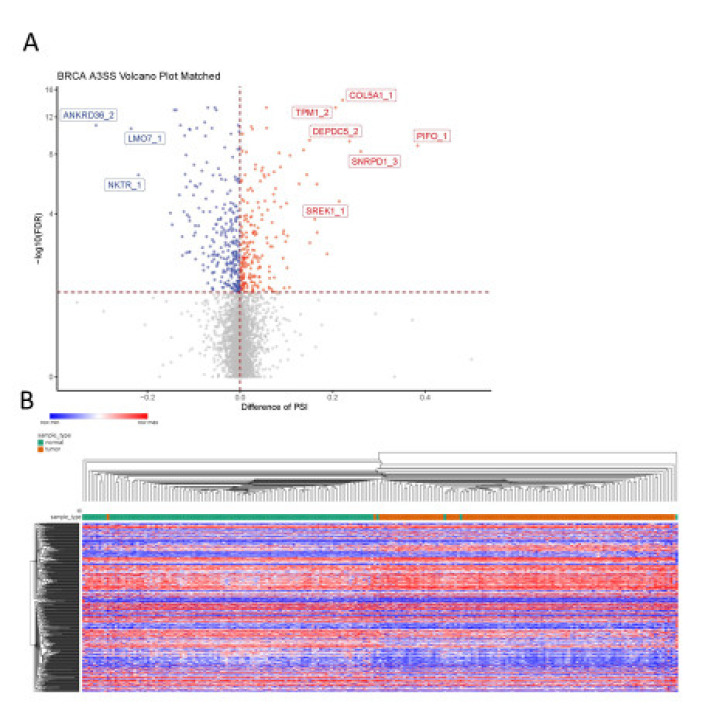
Identification of A3SS events in breast cancer. (**A**) Differences in A3SS events between paired BRCA tumor tissue and adjacent normal tissue; blue indicates significantly decreased PSI in tumor (dPSI < 0, FDR < 0.05) and red indicates significantly increased PSI in tumor (dPSI > 0, FDR < 0.05). (**B**) Heat map shows splicing of significant A3SS events in both tumor samples (orange) and adjacent normal samples (green). The gradual change of color from blue to red represents change in PSI from low to high.

**Figure 8 life-11-00746-f008:**
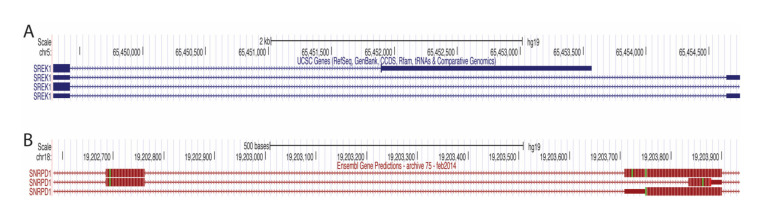
A3SS visualized on the UCSC Genome Browser. (**A**) SREK1 A3SS visualized on the UCSC Genome Browser. Transcripts using proximal 3′ss (top) and distal 3′ss (middle and bottom) located on chromosome 5. Lines with arrows represent the intron. (**B**) SNRPD1 A3SS visualized on the UCSC Genome Browser. Transcripts using proximal 3′ss (top and bottom) and distal 3′ss (middle located on chromosome 18. Lines with arrows represent the intron. The thick rectangles represent the protein-coding region of the exons. The thin rectangles represent the non-protein-coding regions of the exons.

**Figure 9 life-11-00746-f009:**
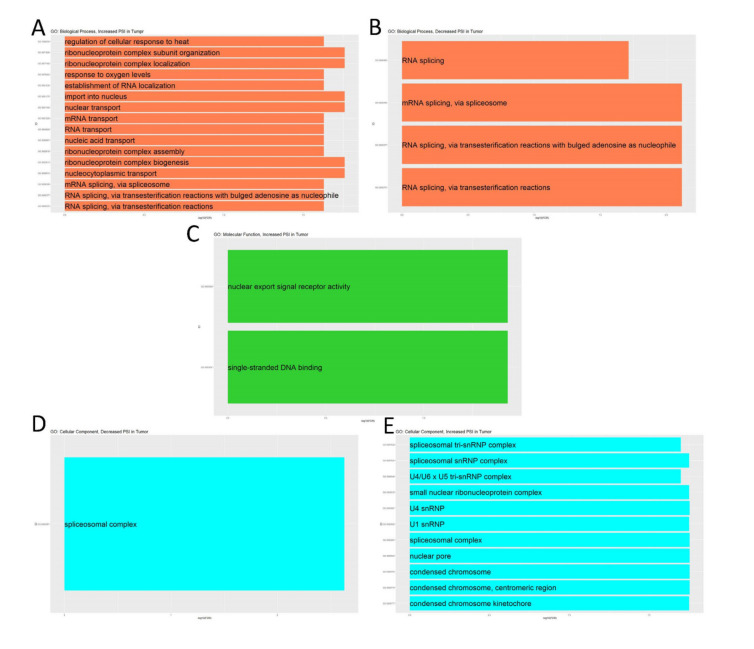
Enrichment analysis of the parent genes of significant A3SS events. Up to the top ten significantly enriched terms (FDR < 0.05) for (**A**,**B**) biological process, (**C**) molecular function, and (**D**,**E**) cellular function. (**A**,**C**,**D**) Analyzed genes that had increased PSI in tumor. (**B**,**E**) Analyzed genes that had decreased PSI in tumor.

**Table 1 life-11-00746-t001:** CXCR3 Splicing Results.

Cohort	Cox Survival *p*-Value	Unpaired Tumor vs. Normal *p*-Value
BRCA	0.64	Not Enough Samples

Not enough samples for survival: <5 tumor samples following filtering (sum of ≥20 reads across both junctions). Not enough samples for tumor vs. normal: <5 tumor sample and <5 normal samples following filtering (sum of ≥20 reads across both junctions).

**Table 2 life-11-00746-t002:** CXCR3 Gene Expression Results.

Cohort	Cox Survival *p*-Value	Tumor vs. Normal *p*-Value
BRCA	0.04	<0.0001

The BRCA cohort had significantly different CXCR3 expression between the tumor and adjacent normal tissue, showing CXCR3 upregulation in the tumor.

## Data Availability

Access to the data presented in this study is available in the Materials and Methods section and in the Appendix A section.

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
