# Peer review of "CXCR3 Expression and Genome-Wide 3′ Splice Site Selection in the TCGA Breast Cancer Cohort"

_life, 2021, doi:10.3390/life11080746_

Round 1
Reviewer 1 Report
well written and well presented paper. Although it corresponds, mainly, to oncologists, it has learning points to the average physician. I suggest you to accept it.
Author Response
We thank Reviewer 1 for the feedback.
Reviewer 2 Report
The authors performed series of meta-analysis regarding CXCR gene expression using TCGA data with breast cancer cohort. The authors identified that CXCR expression is higher in tumor proximity, but interestingly, higher CXCR expression is associated with higher survival. Moreover, authors linked high CXCR3 expression to abundance of immune cell infiltration, which partially explained the previous findings. Although author did not draw clear relationship between BRCA outcome and CXCR isoform expression, the study overall is a good call for attention to the alternative splicing, especially A3SS events, in future gene meta-analysis studies.
I recommend following minor revision before publication:
- Figure issue: Due to reading conventions of most readers, it would be beneficial to place figure letters (A,B,C…) to upper left of panels, instead of upper right.
- Line 206: To accommodate a broad spectrum of readers, please briefly explain subtypes of BRCA to readers. Here, Luminal A breast cancer includes PR and ER positive but not HER2 negative. Also please mention what is luminal B at line 246 as well.
- In figure 9, I understand that GO generated graphs are not ideally formatted and decorated, and PDF compression may yield low quality figures. However, I cannot read anything from the figure clearly. Please enhance the figure to make texts bigger.
Author Response
We thank Reviewer 2 for the feedback. With the improved figures and other noted changes, the text now provides sufficient background about breast cancer subtypes.
- Figure issue: Due to reading conventions of most readers, it would be beneficial to place figure letters (A,B,C…) to upper left of panels, instead of upper right.
Response: We have placed the figure letters on the upper left of panels as indicated by Reviewer #2.
- Line 206: To accommodate a broad spectrum of readers, please briefly explain subtypes of BRCA to readers. Here, Luminal A breast cancer includes PR and ER positive but not HER2 negative. Also please mention what is luminal B at line 246 as well.
Response: We have now included an explanation of subtypes of BRCA and a citation #29 (PNAS, 2001 Sep 11;98(19):10869-74.) to readers in after lines 206 and 246, now lines 211 and 254.
- In figure 9, I understand that GO generated graphs are not ideally formatted and decorated, and PDF compression may yield low quality figures. However, I cannot read anything from the figure clearly. Please enhance the figure to make texts bigger.
Response: We thank Reviewer 2 for the feedback and constructive suggestions. We have re-done the figure so that the text is bigger for improved legibility.